# Eco-Friendly Sanitization of Indoor Environments: Effectiveness of Thyme Essential Oil in Controlling Bioaerosol Levels and Disinfecting Surfaces

**DOI:** 10.3390/biotech13020012

**Published:** 2024-04-26

**Authors:** Daniela Sateriale, Giuseppina Forgione, Giuseppa Anna De Cristofaro, Leonardo Continisio, Chiara Pagliuca, Roberta Colicchio, Paola Salvatore, Marina Paolucci, Caterina Pagliarulo

**Affiliations:** 1Department of Science and Technology, University of Sannio, Via F. De Sanctis Snc, 82100 Benevento, Italy; sateriale@unisannio.it (D.S.); gforgione@unisannio.it (G.F.); gadecristofaro@unisannio.it (G.A.D.C.); paolucci@unisannio.it (M.P.); 2Department of Molecular Medicine and Medical Biotechnologies, University of Naples Federico II, Via S. Pansini 5, 80131 Naples, Italy; leonardo.continisio01@universitadipavia.it (L.C.); chiara.pagliuca@unina.it (C.P.); roberta.colicchio@unina.it (R.C.); psalvato@unina.it (P.S.); 3Department of Public Health, Experimental and Forensic Medicine, University of Pavia, Viale Golgi 19, 27100 Pavia, Italy; 4CEINGE-Biotecnologie Avanzate s.c.ar.l., Via G. Salvatore 486, 80145 Naples, Italy

**Keywords:** indoor sanitization, thyme essential oil, antimicrobial activity, bioaerosol control, surface disinfection

## Abstract

Bioaerosols and pathogens in indoor workplaces and residential environments are the primary culprits of several infections. Techniques for sanitizing air and surfaces typically involve the use of UV rays or chemical sanitizers, which may release chemical residues harmful to human health. Essential oils, natural substances derived from plants, which exhibit broad antimicrobial properties, could be a viable alternative for air and surface sanitation. The objective of this study has been to investigate the efficacy of thyme essential oil (TEO) in environmental sanitation processes. In Vitro assays through agar well diffusion, disk volatilization and tube dilution methods revealed significant antimicrobial activity of TEO 100% against foodborne and environmental isolates, with both bacteriostatic/fungistatic and bactericidal/fungicidal effects. Therefore, aqueous solutions of TEO 2.5% and 5% were formulated for air sanitation through nebulization and surface disinfection via direct contact. Bioaerosol samples and surface swabs were analyzed before and after sanitation, demonstrating the efficacy of aqueous solutions of TEO in reducing mesophilic and psychrophilic bacteria and environmental fungi levels in both air and on surfaces. The obtained results prove the antimicrobial potential of aqueous solutions of TEO in improving indoor air quality and surface cleanliness, suggesting thyme essential oil as an effective and safe natural sanitizer with minimal environmental impact compared to dangerous chemical disinfectants.

## 1. Introduction

In recent decades, research in the field of environmental sanitation has received increasing attention, primarily linked to the awareness of the risks associated with the microbial contamination of enclosed environments and even more so after the recent SARS-CoV-2 pandemic. In response to this demand, there has been a notable surge of interest in alternative approaches to disinfection that prioritize effectiveness, safety and sustainability. Natural compounds, derived either from plants or from the processing byproducts of the agrifood industry, have garnered significant economic attention owing to their attributes. Their robust antioxidant, antibacterial and antifungal properties make them suitable for various industrial sectors, such as food, nutraceutical and pharmaceutical ones [1,2,3,4]. In particular, the use of essential oils as antimicrobial agents arouses more interest due to their unique biological properties and their potential for limited environmental impact. Essential oils, extracted from aromatic plants, have been used for centuries for their therapeutic and anti-inflammatory properties [5]. However, only recently their efficacy as antimicrobial agents has been the subject of in-depth scientific investigation. Due to their abundance in various active components like terpenes, terpenoids, carotenoids, coumarins and curcumins, essential oils play a significant role in science offering potent antimicrobial, anti-inflammatory, antioxidant and beneficial properties [6]. Owing to their diverse attributes, essential oils can be used in a natural, safe, environmentally friendly, cost-effective and sustainable manner [7]. They can be used both for food storage [8] and for the improvement of indoor air quality and surface sanitization protocols [9]. Bioaerosol could potentially make a significant impact on human health; surfaces may become contaminated by airborne microorganisms, and humans can either inhale pathogens or introduce them through food consumption. Effective methods for minimizing the risk of infection from direct exposure to airborne pathogens include air filtration, exposure to UV light and the deactivation of bioaerosols using chemical reagents [10]. In particular, the common use of chemical disinfectants has the potential to trigger complex reactions, leading to the release of reactive intermediates and byproducts into the indoor environment. Continuous exposure to both primary and secondary products from chemical cleaners may pose a significant exposure risk, with side effects on human health [11]. In this context, a new method for environmental sanitization could be the use of natural substances with antimicrobial activity, without releasing harmful residues for human health, such as plant essential oils.

Essential oils are obtained from plants through water vapor distillation, dry distillation or mechanical extraction [9]. These oils contain over 100 different chemical compounds, most of which are odorant and include monoterpenes, sesquiterpenes, terpenes and terpenoids [6]. Their diverse chemical composition gives them a broad spectrum of antimicrobial activity against bacteria, viruses and fungi. In fact, depending on their composition, essential oils can inhibit the metabolic functions of a wide range of microorganisms. Several studies demonstrated that cinnamon, lemongrass and thyme essential oils, when in gaseous contact, inhibit the growth of microorganisms responsible for respiratory infections [12]. Thyme essential oil was shown to possess strong antibacterial activity against *Salmonella typhi*, *Staphylococcus aureus* and *Pseudomonas aeruginosa* [13]. Thyme essential oil is known for its potent antibacterial, antiviral and antifungal properties, mainly attributed to the presence of compounds such as thymol and carvacrol [14]. Thyme essential oil has also demonstrated inhibitory effects on the growth of food-contaminating microorganisms [15], as well as exhibiting anti-biofilm activity against *S. typhimurium* and *B. cereus* [16].

This paper aims to examine the role of TEO in environmental sanitation, initially evaluating its in vitro antimicrobial activity against foodborne pathogens and microorganisms present in bioaerosols, including bacteria, molds and yeasts. Subsequently, we prepared solutions of TEO at different concentrations to also test the ability to sanitize the indoor air using an atomizer and household surface sanitization such as stainless steel and stoneware. The objective was to demonstrate the effectiveness of TEO to improve the microbiological quality of indoor air and surface hygiene through the reduction in the total microbial population and potential human pathogens.

## 2. Materials and Methods

### 2.1. Sanitizing Agents

At the Microbiology Laboratory of the Department of Sciences and Technologies (DST) of the University of Sannio, the antimicrobial properties of TEO (thyme oil- white FG, W306509, Sigma Aldrich, Merck, Darmstadt, Germany) and of two TEO sanitizing solutions were evaluated. The sanitizing formulations consisted of TEO at percentages of 2.5% (*v*/*v*) and 5% (*v*/*v*) in distilled water. Both formulations needed vigorous shaking before each use to ensure homogeneous distribution of TEO micelle aggregates in solution. TEO was chosen as it is approved by the EFSA (European Food Safety Authority) for food use, as it has broad-spectrum antimicrobial activity in small quantities and does not release toxic residues but has a pleasant odor that can be exploited for aromatherapy.

### 2.2. Microbial Strains

The microbial strains chosen for the in vitro antimicrobial tests are American Type Culture Collection (ATCC) food pathogen strains, generously provided by Istituto Zooprofilattico Sperimentale del Mezzogiorno (IZSM) site in Portici (NA, Italy), including *E. coli* ATCC 25922, *S. aureus* ATCC 25923, *S. enterica* ATCC 14028 and *B. cereus* ATCC 14579. Furthermore, as representatives of the bioaerosol microorganisms, we selected environment isolates, provided by the Department of Molecular Medicine and Medical Biotechnologies and the Department of Agriculture of the University of Naples Federico II, including *P. aeruginosa*, *P. fluorescens*, *R. radiobacter*, *Y. enterocolitica*, *C. albicans* and *A. flavus*.

### 2.3. Preliminary Antimicrobial Assays

#### 2.3.1. Agar Well Diffusion Method

To qualitatively assess the antimicrobial activity of TEO against tested foodborne and environmental isolates, a preliminary in vitro antimicrobial screening was performed through the standardized agar well diffusion assay described by Perez et al. [17], with a few changes. In brief, aliquots of 200 µL of each microbial suspension, with an adjusted optical density (O.D.) of 0.5 at 600 nm, were spread on Petri dishes of Luria Bertani (LB) agar. Then, wells with a diameter of 6 mm, appropriately spaced, were created using sterile glass Pasteur pipettes (Sigma-Aldrich S.r.l., Milano, Italy). Aliquots of TEO 100% (10, 20 and 40 µL) were loaded into the wells. After incubation under aerobic conditions at 37 ± 2 °C for 24–48 h for all bacterial strains and *C. albicans*, and at 25 ± 2 °C for 48–72 h for *A. flavus*, the diameter (expressed in mm) of the inhibition zones formed around the wells was measured [18]. Antimicrobial activities were expressed as the mean diameter of the inhibition zones (MDIZ) produced by the thyme essential oil against the tested microorganisms. Common chemical sanitizers including NaClO 3% (Candidalba, San Marco Evangelista, Caserta, Italy) and H_2_O_2_ 3% (BioXCare, Euthalia, Rome, Italy) and conventional antimicrobials including gentamicin (Sigma-Aldrich S.r.l., Milano, Italy), vancomycin (Gold-biotechnology, Saint Louis, MI, USA), amoxicillin (Aesculapius Farmaceutici S.r.l., Brescia, Italy) and tioconazole (Sigma-Aldrich S.r.l., Milano, Italy) were used as positive controls, while sterile distilled water was used as the negative control for the assay. The experiments were conducted in triplicate with independent cultures.

#### 2.3.2. Disk Volatilization Method

To analyze the antimicrobial activity of the volatile compounds of TEO 100%, the disk volatilization method was carried out, as described by Sateriale et al. [15]. Briefly, after the distribution of 200 µL of each microbial suspension on LB agar in Petri dishes, filter paper discs loaded with TEO (10 µL) were placed at the center of the Petri dish lid, instead of on the solid medium to avoid the direct contact of TEO with the medium. The plates were inverted to allow volatile compounds to vaporize and exert their antimicrobial effect on the pure culture setup. NaClO 3% and H_2_O_2_ 3% were used as a positive control. The plates were then incubated in appropriate growth conditions. After growth, the diameter (expressed in mm) of the inhibition zones was measured, and antimicrobial activity was expressed as the mean diameter of the inhibition zones (MDIZ) produced by the volatile compounds of the TEO against the tested microorganisms.

#### 2.3.3. Tube Dilution Method

To calculate the minimum inhibitory concentration (MIC) and minimum bactericidal/fungicidal concentration (MBC/MFC), scalar decreasing dilutions of TEO 100% were prepared in LB broth. These dilutions were then incubated for 24 h at the growth temperature of microorganisms test with standard microbial inoculum 1 × 10^5^ CFU/mL (colony-forming units/mL) according to Clinical and Laboratory Standards Institute (CLSI) 2022 guidelines [18], in order to quantitatively determine the susceptibility of foodborne and environmental isolates to different final concentrations of TEO (0, 0.1, 0.2, 0.4, 0.5, 0.8, 1, 1.5, 2, 2.5, 4, 5, 10, 15, 20, 25, 40, 50, 100, 150, 200, 250 μL mL^−1^). After incubation, MIC values were determined by evaluating the turbidity of the tubes. The aliquots of each test tube were spread on agar (Luria Bertani for bacterial strains and Sabouraud dextrose agar for *C. albicans* and *A. flavus*) and incubated under conditions necessary for microbial growth. After a second incubation under optimal growth conditions, the growth of viable colonies was observed, to confirm the MIC and determine MBC. In particular, MIC has been assigned to the lowest concentration of antimicrobial agent capable of preventing 50% of microbial growth, while MBC was assigned to the lowest concentration of antimicrobial agent that can kill 99% of microorganisms.

### 2.4. Experimental Sanitization Procedure Location

Air sampling was conducted in the Microbiology Laboratory of the Department of Sciences and Technologies of the University of Sannio (Benevento, Italy). The environment measures 49 m^2^ and is schematically represented in Figure 1. The sanitization was carried out with the lights turned off and the ventilation and heating system turned off to avoid changes and movement of the air. During sampling, the temperature was approximately 25 ± 2 °C, and the humidity was below 50%; the lights and the ventilation system were off. Throughout the sampling procedure, no staff members were present, and the environment was unoccupied.

### 2.5. Sanitization of Indoor Environment through Nebulization

The indoor environment sanitization was performed by dispensing the sanitizing TEO-based solution using an atomizer (X-POWER F-16, XPOWER Manufacture Inc., City of Industry, CA, USA). Due to the pressure applied inside the atomizer, the liquid is nebulized into atomized particles (sizes ranging < 50 μm, 50 mL min^−1^), ensuring uniform sanitization of the treated environment. A total quantity of 100 mL of two experimental sanitizing formulations, TEO 2.5% and 5%, was used, based on the volume of the indoor environment to be sanitized. In particular, the time to nebulize 100 mL was 3 min, and the sampling was performed immediately after sanitization. Each sanitization intervention was conducted at weekly intervals to allow environmental microbial repopulation. As a control, 0.5% hydrogen peroxide in stabilized solution was used, commonly employed as sanitizing solution in line with the directives for sanitizing indoor work environments, including healthcare settings, as per the Italian Ministry of Health guidelines for indoor environment sanitization (Circular 5443 dated 22 February 2020). Hydrogen peroxide solution was dispersed at the same volume and time.

### 2.6. Indoor Bioaerosol Sampling

For the monitoring analysis of the microbiological air quality in indoor environment, the passive bioaerosol sampling method was selected. This method involves a quantitative estimation of air health levels by exposing plates containing suitable culture media in the examined environment for specified time intervals of 1 h. Microorganisms carried by solid or liquid particles suspended in the air are collected through sedimentation on plates containing selected media suitable for the growth of targeted microorganisms in appropriate growth conditions (Luria Bertani agar for total mesophilic and total psychrophilic bacteria, at 37 ± 2 °C and 18 ± 2 °C, respectively; Sabouraud dextrose agar for total fungi at 25 ± 2 °C). In particular, sampling was carried out by placing media in Petri dishes at multiple sampling points within the examined environment. The selection of sampling points was based on the size and characteristics of the chosen environment, always maintaining a distance of at least 1 m from the ground and any internal furnishings and away from ventilation systems and windows. For all sampling tests, the plates were positioned at 3 different points in the room, as indicated by yellow asterisks in Figure 1. After each sanitization, the plates were consistently placed in the same location to avoid position-dependent variations. Sampling was conducted both before and after sanitization, with triplicate measurements taken for each sampling point and type of media. At the end of the sampling, the plates were immediately incubated under appropriate growth conditions, and then the colony-forming units (CFUs) were counted.

### 2.7. Indoor Environmental Surface Sanitization

Surfaces were sanitized by applying the sanitizer onto the surface (a square of 10 × 10 cm) using a spray dispenser, and, after an action time of about 10 s, the product was uniformly dispersed on the surface with paper towels. Before sampling, the surface was allowed to dry for a few seconds. The dose of sanitizing solution used was 250 µL for each type of sanitizer, including the aqueous solution of TEO 2.5% and 5% and the solution of NaClO 3% chosen as positive control. The sanitization procedure was conducted on two types of surfaces, stainless steel and stoneware.

### 2.8. Indoor Environmental Surface Sampling

To evaluate the effectiveness of sanitization procedures, the extent of contamination on environmental surfaces was estimated. For sampling, swabs moistened in a sterile isotonic solution of sodium chloride (0.85% NaCl, Merck, Darmstadt, Germany) were used, in accordance with Centers for Disease Control and Prevention guidelines [19]. Each swab was rubbed on defined surfaces measuring 10 × 10 cm along trajectories covering the entire surface to be analyzed, with cross-diagonal movements, 10 times in each direction. Sampling before and after sanitization was conducted in adjacent areas of each of the previously indicated sizes. Immediately after swabbing, the samples were streaked on the different culture media and incubated in appropriate growth conditions, in detail Luria Bertani agar for total mesophilic and total psychrophilic bacteria, at 37 ± 2 °C and 18 ± 2 °C, respectively, and Sabouraud dextrose agar for total fungi at 25 ± 2 °C. After incubation, the counting of CFUs was performed. Results were expressed as number of CFUs per cm^2^ of examined surface. The microbial sampling and viable count were conducted on each type of selected surface both before and after sanitization.

## 3. Results

### 3.1. In Vitro Antimicrobial Activity of TEO 100%

TEO 100% showed significant in vitro antimicrobial activity against tested microorganisms.

Table 1 and Appendix A show the mean diameters of the inhibition zones (MDIZ) estimated by the agar well diffusion method for TEO and positive controls against the selected bacterial and fungal strains. TEO exhibited appreciable inhibitory activity against all tested foodborne bacterial strains (*E. coli* ATCC 25922, *S. aureus* ATCC 25923, *S. enterica* ATCC 14028 and *B. cereus* ATCC 14579), as confirmed by the values of inhibition zones of bacterial growth. The MDIZ reached values up to 33.00 ± 1.41 mm (vs. *E. coli* ATCC 25922), 31.50 ± 3.54 mm (vs. *S. aureus* ATCC 25923), 27.00 ± 0.00 mm (vs. *S. enterica* ATCC 14028) and 27.00 ± 0.00 mm (vs. *B. cereus* ATCC 14579) when TEO 100% was tested at the volume of 40 µL/well. TEO showed significant antimicrobial effects also against environmental isolates, both bacterial (*P. aeruginosa*, *P. fluorescens*, *R. radiobacter*, *Y. enterocolitica*) and fungal (*C. albicans* and *A. flavus*) ones, with MDIZ ranging between 9.00 ± 1.41 mm (TEO 100% 10 µL/well vs. *R. radiobacter*) and 36.50 ± 2.12 mm (TEO 100% 40 µL/well vs. *Y. enterocolitica*). The positive controls, including gentamicin, vancomycin, amoxicillin, tioconazole, NaClO 3% and H_2_O_2_ 3%, demonstrated antimicrobial effects against all microbial strains. For the resistance/sensitivity cutoff values of the associations between test microorganisms and antibiotics used as positive controls, please refer to the European Committee on Antimicrobial Susceptibility Testing (EUCAST) tables (https://www.eucast.org/clinical_breakpoints, accessed on 9 February 2024; https://www.eucast.org/fileadmin/src/media/PDFs/EUCAST_files/Breakpoint_tables/v_14.0_Breakpoint_Tables.pdf, accessed on 9 February 2024). In addition, NaClO 3% and H_2_O_2_ 3% solutions showed antimicrobial effects against the tested microorganisms, apart from NaClO 3% against *B. cereus* ATCC 14579 for which no inhibition zones were detected; no effects were observed for the negative control.

The disk volatilization method allowed for the evaluation of the antimicrobial effects of the volatile components contained in TEO 100%. The results from this test (Table 2 and Appendix A) demonstrated the high antimicrobial effects of TEO against both foodborne and environmental isolates when tested in the vapor phase. The MDIZ of microbial growth reached values up to 40.50 ± 2.12 mm (TEO 100% vs. *E. coli* ATCC 25922) against the tested foodborne pathogens and up to 60.00 ± 0.00 mm (TEO 100% vs. *R. radiobacter*) against environmental isolates.

The in vitro antimicrobial properties of TEO 100% were confirmed also by quantitative assays. The MIC and the MBC/MFC values of TEO 100% and positive controls (NaClO 3%, H_2_O_2_ 3%, gentamicin, vancomycin, amoxicillin and tioconazole) are reported in Table 3. Appendix A shows images of the in vitro antimicrobial activity of the tested antimicrobial agents detected by quantitative assays. EUCAST tables with cutoff values of antibiotics/test microorganisms associations are available at https://www.eucast.org/clinical_breakpoints (accessed on 9 February 2024) and https://www.eucast.org/fileadmin/src/media/PDFs/EUCAST_files/Breakpoint_tables/v_14.0_Breakpoint_Tables.pdf (accessed on 9 February 2024). The MIC values of TEO 100% ranged between 0.1 and 0.5 µL mL^−1^, except for *P. aeruginosa* (20 µL mL^−1^) and *B. cereus* (150 µL mL^−1^). The MIC values of NaClO 3% ranged between 2 µL mL^−1^ and 40 µL mL^−1^, except for *B. cereus* (150 µL mL^−1^), confirming the resistance shown in previous assays. The MIC values of H_2_O_2_ ranged between 1.5 µL mL^−1^ and 20 µL mL^−1^. The MBC/MFC values of TEO 100% ranged between 0.1 and 1.5 µL mL^−1^, significantly lower than positive controls, except for *P. aeruginosa* and *B. cereus*, which are less sensitive to TEO 100%, with MBC values of 80 µL mL^−1^ and 250 µL mL^−1^, respectively. These results demonstrated TEO 100% bacteriostatic and bactericidal effects against all tested foodborne and environmental bacterial isolates. Regarding fungistatic and fungicidal effects of TEO 100%, MIC and MFC values against *C. albicans* are 0.2 µL mL^−1^ and 0.8 µL mL^−1^, while for *A. flavus*, due to the organism’s type and its own physiology, the data were not determined.

### 3.2. Indoor Sanitization

#### 3.2.1. Indoor Air Sanitization with TEO 2.5% and TEO 5% Aqueous Solutions

Air sanitization was performed using an atomizer, as indicated in the Materials and Methods (Section 2.5), at the Microbiology Laboratory of the University of Sannio. Figure 2 shows the total microbial levels of microbiological air monitoring before and after sanitization procedures with aqueous solutions of TEO 2.5% and 5% and a stabilized solution of H_2_O_2_ 0.5%. Analyzing the total mesophilic, psychrophilic and fungal populations present in the air of the sampled environment, it is observed that the sanitizing solution with 2.5% TEO exerted excellent antibacterial and antifungal action, leading to a halving of the total count compared to the control sampling (Figure 2). The growth of total mesophilic bacteria decreased from 5.44 ± 2.12 to 2.67 ± 1.87, while the count of total psychrophilic bacteria decreased from 2.89 ± 1.62 to 1.78 ± 1.20; finally, molds and yeasts decreased from 1.33 ± 1.12 to 0.56 ± 0.73. The results obtained after the nebulization of the 5% TEO solution are even more significant. The total fungal count is equal to 0, with fungicidal effects superior to H_2_O_2_ 0.5%. The mesophilic and psychrophilic bacterial counts are statistically reduced, respectively, to 0.78 ± 0.37 and 0.33 ± 0.50, with bactericidal effects comparable to those obtained with hydrogen peroxide (Figure 2).

#### 3.2.2. Surface Sanitization with TEO 2.5% and TEO 5% Aqueous Solutions

The sanitization of surfaces was carried out on stainless-steel and monolithic stoneware shelves at the Microbiology Laboratory of the University of Sannio. The sanitizers used for surface sanitization included aqueous solutions of TEO 2.5% and 5% as an alternative to NaClO 3%, commonly used to disinfect laboratory surfaces, used as a positive control. Sampling procedures by sterile swabs were carried out on both surface types in triplicate before and after each sanitization. Figure 3 shows the total counts of mesophilic and psychrophilic bacteria and fungal populations found after each sampling. The results demonstrate that the TEO solutions significantly reduced the total microbial population. In particular, it was observed that the TEO 2.5% eliminated all microorganisms when used on the stainless-steel shelf (Figure 3a), while on the monolithic stoneware, there was a significant reduction but not complete absence of microorganisms (Figure 3b). On the other hand, the TEO 5% exerted an antibacterial and antifungal action comparable to NaClO 3% on both analyzed surfaces (Figure 3a,b).

## 4. Discussion

TEO has gained attention for its rich composition of bioactive compounds, particularly thymol, which exhibit strong antimicrobial properties [20]. Our findings corroborate previous research demonstrating the efficacy of TEO against a wide range of microbial pathogens, including foodborne and environmental bacterial and fungal isolates. *Escherichia coli*, *Salmonella enterica*, *Staphylococcus aureus* and *Bacillus cereus* are foodborne bacteria of great concern for foodborne illness outbreaks [21]; they are among the most common etiological agents of human gastroenteritis and cause serious food safety issues for the food industry due to the formation of biofilms, spores and/or emetic toxins [22]. Environmental pathogens have a significant impact on human health as well. These microorganisms, including bacteria and fungi, thrive in various environments, from water to soil sources (*Pseudomonaceae*, *Candida* spp., *Rhizobium radiobacter*, etc.), and can contaminate food supplies (*Yiersinia enterocolitica*) [23]. Their presence contributes to the spread of different diseases; they can persist and adapt to changing conditions, causing infections difficult to control and eradicate with standard therapies and procedures.

Analyzing the individual tested bacterial strains, TEO 100% exhibited significant antibacterial activity against *E. coli* ATCC 25922 food isolate, with values of the MDIZ significantly higher with respect to the gentamicin and comparable to the inhibition zone observed for NaClO 3% at the volume of 40 µL. *S. enterica* ATCC 14028 food isolate was shown to be sensitive to TEO already at the volume of 10 µL; TEO 100% significantly reduced the growth of *S. enterica* ATCC 14028 also when tested at the volume of 20 µL, showing antimicrobial activity comparable to H_2_O_2_ 3% at 40 µL. Regarding Gram-positive bacterial foodborne strains, TEO 100% showed significant antimicrobial activity against *S. aureus* ATCC 25923 and *B. cereus* ATCC 14579 when used at 10 µL, with MDIZ values comparable and/or significantly higher to positive controls. Concerning environmental isolates, *P. aeruginosa* was shown to be more sensitive to TEO 100% with respect to *P. fluorescens*. *R. radiobacter* was shown to be the most resistant bacterial isolate to TEO 100%, when tested in the liquid phase, with MDIZ values significantly lower with respect to controls, while the antimicrobial activity of TEO 100% against *Y. enterocolitica* was shown to be comparable to or higher than the control antibiotic, at increasing volumes. The results from the disk volatilization method showed the antibacterial activity of volatile compounds from TEO 100% against both foodborne and environmental bacterial strains. Microbial inhibition zone measurements demonstrate that the volatile compounds of TEO 100% had an antibacterial effect against the tested isolates, in some cases significantly higher than NaClO 3% and H_2_O_2_ 3%. In particular, *S. enterica* ATCC 14028 was shown to be the most resistant foodborne pathogen to TEO 100%, while *E. coli* ATCC 25922 was shown to be the most sensitive one; among environmental isolates, *R. radiobacter*, *Y. enterocolitica*, *P. fluorescens* and *P. aeruginosa* were shown to be significantly sensitive to the volatile compounds of TEO 100%. The in vitro antibacterial properties of TEO 100% were confirmed also by quantitative assays, through the determination of MIC and MBC values, thus demonstrating TEO bacteriostatic and bactericidal effects against all tested foodborne and environmental bacterial isolates, in both liquid and vapor phases. These results are in line with other studies about the antibacterial activity of thyme essential oil [5,24]. They confirm that the use of very low concentrations of TEO 100% is sufficient to inhibit the growth of bacteria present in the air and on inanimate surfaces. The proven antibacterial properties of TEO in both liquid and vapor phases closely depend on its chemical composition. Thymol directly disrupts cellular structure, compromising the integrity of the cell membrane. This is demonstrated by an increase in the permeability of both inner and outer membranes, resulting in the leakage of intracellular biological macromolecules such as DNA. Additionally, the accumulation of intracellular reactive oxygen species (ROS) triggered by TEO and thymol can lead to DNA damage through oxidative stress. Thymol further exacerbates this by intercalating into DNA, disrupting its normal function and ultimately causing bacterial cell death [25]. Even though usually Gram-positive bacteria are more susceptible to essential oils than Gram-negative bacteria [26], we observe that a much lower quantity of TEO 100% is necessary to inhibit the growth of Gram-negative bacteria. In fact, thymol, the major component of thyme essential oil, caused the disruption of the outer membrane of Gram-negative bacteria, releasing lipopolysaccharides (LPS) and increasing the permeability of the cytoplasmic membrane, so exerting very strong antimicrobial activity [27].

Agar diffusion assays also highlighted the antifungal activity of TEO 100% against *C. albicans* and *A. flavus*, with MDIZ values statistically higher compared to tioconazole and to the NaClO 3% control. TEO volatile components also exerted antifungal action against both tested fungi, while the tube dilution assay demonstrated the fungistatic and fungicidal effects of TEO only against *C. albicans*. *A. flavus* was shown to be resistant to the tested concentration of TEO 100%. Comparing these results with the literature, recent studies observed how thyme essential oil is responsible for cellular damage in *Candida* species by forming protuberances and holes in the cell wall [28]. Furthermore, thymol is capable of inhibiting ergosterol biosynthesis and disrupting membrane integrity in fungal cells [29]. It was also observed that, in the vapor phase, essential oil impact molds at various stages of their lifecycle, including germination, hyphal growth and sporulation [30].

Given the significant results of the in vitro assays conducted, we proceeded to analyze the antibacterial and antifungal activity of TEO when incorporated into aqueous solutions used for environmental sanitization, both for surfaces and air biocontrol. Although there are many studies on the use of thyme oil as a food preservative and in aromatherapy practices, there is still a lack of literature on its use as an environmental sanitizer; this underlines the importance and novelty of this study.

Regarding surface sanitization, TEO solutions were demonstrated to significantly reduce the microbial charge on sampled surfaces. In particular, TEO solutions demonstrated higher efficacy on stainless steel surfaces compared to monolithic stoneware when used at a concentration of 2.5%, while there is complete eradication of all microorganisms on both surfaces when TEO is used at a concentration of 5%. The different results depend on the type of surface, as the monolithic stoneware proves to be more difficult to clean due to its porous characteristic. It was proven that sanitizers exhibit greater efficacy in disinfecting smooth surfaces, such as steel, compared to rough and porous ones [31]. However, the obtained results demonstrated that TEO solutions can be used as a natural, biocompatible sanitizer, with antimicrobial properties comparable to common chemical sanitizers. According to Gelmini et al. [32], the application of natural solutions for surface disinfection containing essential oils could provide sanitizing effects mostly equivalent to or superior to chemical disinfectants. The tested TEO solution makes the surfaces clean and drastically reduces the total microbial charge, leaving a pleasant odor and shine, especially for stainless steel.

Analyses of bioaerosol levels in the samples of the indoor environment demonstrated the potential use of TEO as an environmentally friendly alternative to common chemical products used for indoor air sanitization. Furthermore, the results obtained are in line with the requirements of the Italian regulations regarding the sanitization of indoor environments. The obtained results showed that TEO solutions can cause a drastic lowering of the microbial charge already if used at 2.5% and a total reset at 5%, thus demonstrating that with a minimal concentration of essential oil, it is possible to drastically reduce or even eliminate the presence of microorganisms typically found in bioaerosols. Other studies demonstrated that EO vapors have an effect on the microbial count in bioaerosols by reducing the bacterial levels only to a certain extent (43%) but not wholly eradicating the bacteria [33]. In our study, microbial counts reached levels comparable to those achieved through standard sanitization using a stabilized hydrogen peroxide solution (H_2_O_2_ 0.5%). Such sanitization could be considered environmentally friendly as it does not result in the release of high amounts of residues considered toxic, unlike NaClO [25,28].

One of the primary advantages of utilizing TEO lies in its natural origin, positioning it as a promising alternative to synthetic antimicrobial agents commonly employed in disinfection processes. As concerns regarding the environmental impact of chemical disinfectants continue to escalate, there is a growing demand for eco-friendly alternatives such as TEO. Like other plant-derived essential oils, thyme essential oil is biodegradable and renewable, aligning with environmentally friendly practices in sanitation. This enhances its appeal as a sustainable solution for environmental sanitation. In addition, thyme essential oil has a pleasant herbal scent that can contribute to a more appealing olfactory environment during sanitization procedures, unlike some harsh chemical disinfectants, such as sodium hypochlorite [34,35]. In addition, several studies have evaluated the potential therapeutic uses of thyme essential oil for the treatment of disorders affecting the respiratory, nervous and cardiovascular systems, promoting its use in working environments [36].

Despite the significant biological benefits related to the use of TEO, several potential risks associated with its use in indoor environments are still questioned. Some in vitro and in vivo studies and several case reports have indicated that carvacrol, thymol and eugenol may have potential toxicological effects after direct contact [37]. Concerning allergenicity, it seems that thymol/thyme may act as an allergen, but it may also attenuate allergic inflammation [36]. Regarding TEO effects in the vapor phase, although the utilization of TEO aligns with the principles of green chemistry, emphasizing the design of chemical products and processes that minimize the use and generation of hazardous substances [38], the diffusion of essential oils in the indoor environment can temporarily increase total terpene concentrations, surpassing critical exposure thresholds for specific terpenes. These compounds may have implications for indoor air quality, potentially affecting it for periods ranging from 6 h to 50 days, depending on the method of diffusion employed [9]. These limitations have been associated only with high concentrations of thyme essential oil. Further studies are required to better define risk scenarios, in order to address human exposure to volatile compounds (VOCs) from TEO and to establish its potential allergenicity and toxicity.

## 5. Conclusions

TEO 100% exhibits broad-spectrum antimicrobial activity against several bacterial and fungal strains found in foods and environments, as demonstrated by the preliminary in vitro antimicrobial assays. When incorporated into aqueous sanitizing solutions and nebulized, TEO can provide an excellent alternative to airborne disinfectants. Furthermore, it can be successfully utilized for surface sanitization, achieving satisfactory outcomes on several types of surfaces. These results demonstrate the potential use of thyme essential oil, due to its antibacterial and antifungal properties, as an eco-friendly alternative to common chemical products used for indoor sanitization.

Although further biosafety studies will be needed, we can conclude that the use of low-concentration solutions of TEO could be recommended as green sanitizers for both domestic and critical environments, such as food industries and hospitals or clinical laboratories.

## Figures and Tables

**Figure 1 biotech-13-00012-f001:**
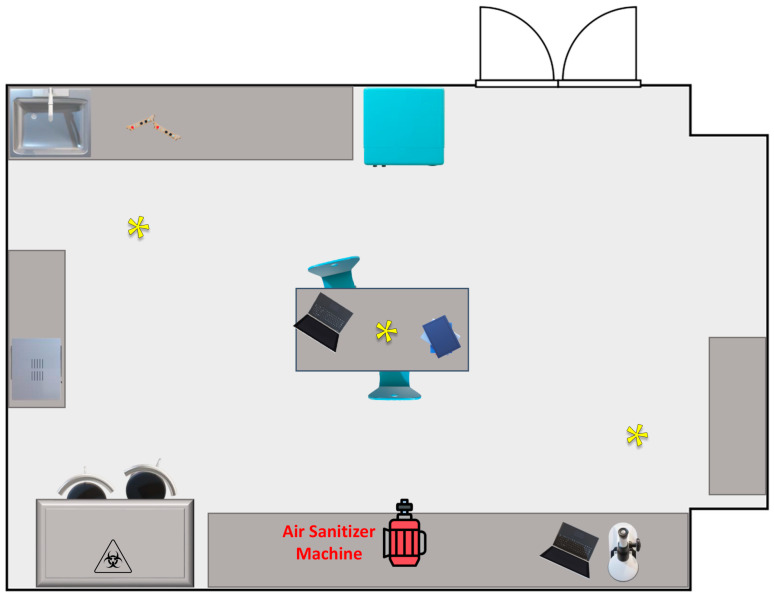
Graphic representation, not to scale, of the indoor environment utilized in the experimental sanitization procedures; asterisks indicate the bioaerosol sampling sites.

**Figure 2 biotech-13-00012-f002:**
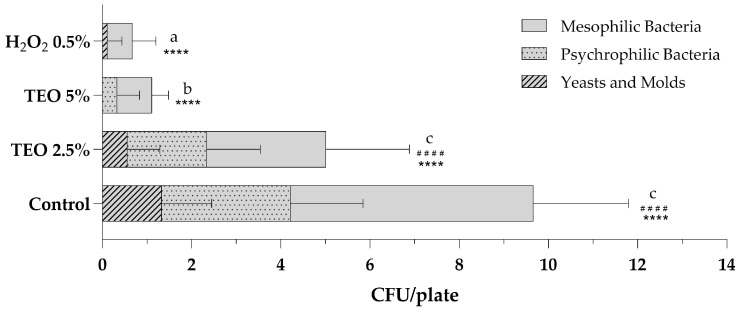
Total microbial levels in indoor bioaerosols before (control) and after sanitization with TEO 2.5% and 5% and H_2_O_2_ 0.5%. The number of colonies for plates is expressed as the mean ± standard deviation of a triplicate assay conducted at three sampling points. TEO, thyme essential oil; H_2_O_2_, hydrogen peroxide. Statistical significance was examined by the two-way ANOVA test with Tukey’s correction (*p* < 0.05) for comparisons with the control. For mesophilic bacteria, asterisks indicate the statistical significance with respect to the positive control (**** *p* < 0.0001); for psychrophilic bacteria, hashtags indicate the statistical significance compared to the control (^####^ *p* < 0.0001); the absence of symbols indicates the absence of significance. Letters (a–c) indicate the statistical differences between different values; results with no significant differences receive the same letter.

**Figure 3 biotech-13-00012-f003:**
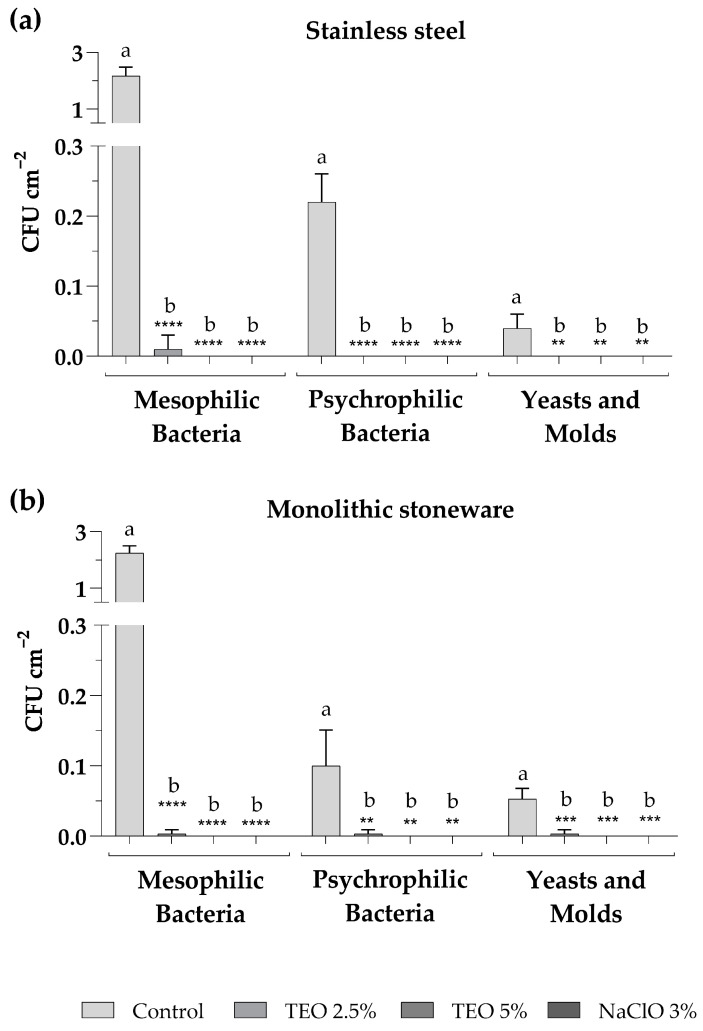
Microbial levels on stainless-steel (**a**) and monolithic stoneware (**b**) surfaces, before (control) and after sanitization with TEO 2.5% and 5% and sodium NaClO 3%. The mean values ± standard deviation, obtained from triplicate analyses, are expressed as colony-forming units (CFU) per 1 square centimeter of sampled surface. Statistical significance was examined by the two-way ANOVA test with Dunnett correction (*p* < 0.05) for comparisons with each control. Asterisks indicate the statistical significance with respect to the positive control (**** *p* < 0.0001; *** *p* < 0.001; ** *p* < 0.01); the absence of asterisks indicates the absence of significance. Letters (a,b) indicate the statistical differences between different values; results with no significant differences receive the same letter.

**Table 1 biotech-13-00012-t001:** In Vitro antibacterial activity of TEO 100% evaluated by agar well diffusion method.

Microorganisms	Volumes Tested	MDIZ (mm)			Positive Control
		TEO 100%	NaClO 3%	H_2_O_2_ 3%	
*E. coli* ATCC 25922	10 µL/well	21.00 ± 2.83 ^a^	13.00 ± 1.41 ^c^ *	19.00 ± 1.41 ^a^	GNT (0.6 mg/well) 19.00 ± 1.41 ^a^
	20 µL/well	30.50 ± 4.95 ^b^ ****	21.00 ± 1.41 ^a^	20.50 ± 0.71 ^a^	
	40 µL/well	33.00 ± 1.41 ^b^ ****	31.00 ± 2.83 ^b^ ****	23.50 ± 0.71 ^a^	
*S. aureus* ATCC 25923	10 µL/well	24.00 ± 1.41 ^a^ *	14.50 ± 0.71 ^c^ *	32.00 ± 1.41 ^b^ ****	VNC (0.4 mg/well) 19.50 ± 0.71 ^a^
	20 µL/well	27.00 ± 0.00 ^a^ ***	20.00 ± 1.41 ^a^	34.50 ± 0.71 ^b^ ****	
	40 µL/well	31.50 ± 3.54 ^b^ ****	31.00 ± 2.83 ^b^ ****	37.50 ± 0.71 ^b^ ****	
*B. cereus* ATCC 14579	10 µL/well	25.50 ± 2.12 ^a^	0.00 ± 0.00 ^b^	11.50 ± 2.12 ^c^	AMX (0.5 mg/well) 29.00 ± 2.83 ^a^
	20 µL/well	27.00 ± 0.00 ^a^	0.00 ± 0.00 ^b^	17.50 ± 0.71 ^d^	
	40 µL/well	27.00 ± 0.00 ^a^	0.00 ± 0.00 ^b^	20.50 ± 0.71 ^e^	
*S. enterica* ATCC 14028	10 µL/well	24.50 ± 3.54 ^a^ ***	11.50 ± 0.71 ^b^	20.50 ± 0.71 ^a^	GNT (0.6 mg/well) 16.00 ± 1.41 ^b^
	20 µL/well	26.00 ± 4.24 ^a^ ****	20.50 ± 0.71 ^a^	23.50 ± 0.71 ^a^ **	
	40 µL/well	29.00 ± 1.41 ^a^ ****	29.50 ± 2.12 ^a^ ****	26.50 ± 0.71 ^a^ ****	
*P. fluorescens*	10 µL/well	17.00 ± 0.00 ^a^ ****	13.50 ± 0.71 ^d^ ****	15.00 ± 0.00 ^d^ ****	GNT (0.25 mg/well) 21.50 ± 0.71 ^b^
	20 µL/well	19.50 ± 0.71 ^b^ *	18.00 ± 1.41 ^a^ ****	23.50 ± 0.71 ^e^ *	
	40 µL/well	30.00 ± 0.00 ^c^ ****	25.50 ± 0.71 ^e^ ****	28.50 ± 0.71 ^c^ ****	
*P. aeruginosa*	10 µL/well	25.50 ± 0.71 ^a^	10.50 ± 3.54 ^b^ ****	12.00 ± 2.83 ^b^ ****	GNT (0.25 mg/well) 28.50 ± 0.71 ^a^
	20 µL/well	28.50 ± 3.54 ^a^	13.00 ± 2.83 ^b^ ***	28.00 ± 1.41 ^a^	
	40 µL/well	31.50 ± 7.78 ^a^	22.50 ± 2.12 ^c^	34.00 ± 1.41 ^a^	
*R. radiobacter*	10 µL/well	9.00 ± 1.41 ^a^ ****	26.00 ± 1.41 ^b^	31.50 ± 0.71 ^e^	GNT (0.25 mg/well) 29.00 ± 0.00 ^b^
	20 µL/well	10.00 ± 0.00 ^a^ ****	39.50 ± 3.54 ^c^ ****	37.00 ± 0.00 ^c^ ***	
	40 µL/well	11.00 ± 0.00 ^a^ ****	59.00 ± 1.41 ^d^ ****	42.50 ± 3.54 ^c^ ****	
*Y. enterocolitica*	10 µL/well	21.50 ± 0.71 ^a^	17.00 ± 4.24 ^a^ ***	16.00 ± 1.41 ^a^	GNT (0.25 mg/well) 27.50 ± 0.71 ^a^
	20 µL/well	30.00 ± 0.00 ^b^	24.00 ± 0.00 ^a^	25.50 ± 4.95 ^a^	
	40 µL/well	36.50 ± 2.12 ^b^ *	34.00 ± 4.24 ^b^ *	34.50 ± 0.71 ^b^ *	
*C. albicans*	10 µL/well	25.00 ± 2.83 ^a^	44.50 ± 0.71 ^c^ ****	10.00 ± 1.41 ^f^ ****	TCZ (0.25 mg/well) 27.50 ± 2.12 ^a^
	20 µL/well	32.50 ± 0.71 ^b^ **	59.50 ± 0.71 ^d^ ****	16.50 ± 0.71 ^g^ ****	
	40 µL/well	34.00 ± 1.41 ^b^ ***	80.50 ± 0.71 ^e^ ****	25.00 ± 1.41 ^a^	
*A. flavus*	10 µL/well	18.50 ± 2.12 ^a^ ***	17.50 ± 0.71 ^a^ ***	11.50 ± 0.71 ^d^ ****	TCZ (0.25 mg/well) 27.00 ± 1.41 ^b^
	20 µL/well	26.00 ± 1.41 ^b^	32.50 ± 0.71 ^b^ **	17.50 ± 3.54 ^a^ ****	
	40 µL/well	31.00 ± 1.41 ^b^	42.00 ± 2.83 ^c^ ****	26.50 ± 0.71 ^b^	

Mean diameter of inhibition zone (in mm) is reported as mean of values obtained from assays in triplicate ± standard deviation. Statistical significance was examined by one-way ANOVA test with Dunnett’s correction (*p* < 0.05) for comparison with positive control and one-way ANOVA test with Tukey’s correction (*p* < 0.05) for multiple comparisons. Asterisks indicate statistical significance with respect to positive control (**** *p* < 0.0001; *** *p* < 0.001; ** *p* < 0.01; * *p* < 0.05); absence of asterisks indicates absence of significance. Letters (a–g) indicate statistical differences between different values; results with no significant differences receive same letter. MDIZ, mean diameter of inhibition zone; TEO, thyme essential oil; NaClO, sodium hypochlorite; H_2_O_2_, hydrogen peroxide; GNT, gentamicin; VNC, vancomycin; AMX, amoxicillin; TCZ, tioconazole. Positive controls used, their concentrations and MDIZ ± standard deviation are reported in column on far right.

**Table 2 biotech-13-00012-t002:** In Vitro antibacterial activity of TEO 100% evaluated by vapor diffusion method.

Microorganisms	MDIZ (mm)		
	TEO 100% (10 µL)	NaClO 3% (10 µL)	H_2_O_2_ 3% (10 µL)
*E. coli* ATCC 25922	40.50 ± 2.12	25.50 ± 0.71 ****	0.00 ± 0.00 ^####^
*S. aureus* ATCC 25923	34.00 ± 1.41	18.00 ± 4.42 ***	26.00 ± 1.41 ^#^
*B. cereus* ATCC 14579	33.50 ± 0.71	0.00 ± 0.00 ****	0.00 ± 0.00 ^####^
*S. enterica* ATCC 14028	26.50 ± 2.12	36.50 ± 2.12 ***	0.00 ± 0.00 ^####^
*P. fluorescens*	45.00 ± 7.07	60.00 ± 0.00 *	10.00 ± 2.83 ^###^
*P. aeruginosa*	41.00 ± 4.24	0.00 ± 0.00 ****	0.00 ± 0.00 ^####^
*R. radiobacter*	60.00 ± 0.00	60.00 ± 0.00	60.00 ± 0.00
*Y. enterocolitica*	55.50 ± 0.71	46.00 ± 1.41 ****	0.00 ± 0.00 ^####^
*C. albicans*	46.50 ± 0.71	49.50 ± 2.12	0.00 ± 0.00 ^####^
*A. flavus*	42.00 ± 0.00	0.00 ± 0.00 ****	0.00 ± 0.00 ^####^

Mean diameter of inhibition zone (in mm) is reported as mean of values obtained from assays in triplicate ± standard deviation. Statistical significance was examined by one-way ANOVA test with Dunnet’s correction (*p* < 0.05) for multiple comparisons. Asterisks indicate statistical significance between TEO 100% and NaClO 3% (**** *p* < 0.0001; *** *p* < 0.001; * *p* < 0.05); hashtags indicate statistical significance between TEO 100% and H_2_O_2_ 3% (^####^ *p* < 0.0001; ^###^ *p* < 0.001; ^#^ *p* < 0.05); absence of asterisks indicates absence of significance. MDIZ, mean diameter of inhibition zone; TEO, thyme essential oil; NaClO, sodium hypochlorite; H_2_O_2_, hydrogen peroxide.

**Table 3 biotech-13-00012-t003:** Quantitative evaluation of in vitro antibacterial activity of thyme essential oil compared to conventional sanitizer.

Microorganisms		TEO 100% [µL mL^−1^]	NaClO 3% [µL mL^−1^]	H_2_O_2_ 3% [µL mL^−1^]	GNT [µg mL^−1^]	VNC [µg mL^−1^]	AMX [µg mL^−1^]	TCZ [µg mL^−1^]
*E. coli* ATCC 25922	MIC	0.2	15	2	4	nt	nt	nt
	MBC	0.5	25	5	10			
*S. aureus* ATCC 25923	MIC	0.5	15	20	nt	1.5	nt	nt
	MBC	1	25	50		2.5		
*B. cereus* ATCC 14579	MIC	150	150	20	nt	nt	50	nt
	MBC	250	250	50			200	
*S. enterica* ATCC 14028	MIC	<0.1	15	20	25	nt	nt	nt
	MBC	0.1	25	40	100			
*P. fluorescens*	MIC	0.4	10	20	0.1	nt	nt	nt
	MBC	1.5	20	40	0.5			
*P. aeruginosa*	MIC	20	40	20	<0.1	nt	nt	nt
	MBC	80	100	100	0.1			
*R. radiobacter*	MIC	0.2	2.5	1.5	<0.1	nt	nt	nt
	MBC	0.8	10	5	0.1			
*Y. enterocolitica*	MIC	0.4	10	10	0.1	nt	nt	nt
	MBC	0.8	20	20	1			
*C. albicans*	MIC	0.2	2	10	nt	nt	nt	100
	MFC	0.8	5	20				250
*A. flavus*	MIC	nd	nd	nd	nt	nt	nt	nd
	MFC	nd	nd	nd				

MIC, minimum inhibitory concentration; MBC, minimum bactericidal concentration; MFC, minimum fungicidal concentration; TEO, thyme essential oil; NaClO, sodium hypochlorite; H_2_O_2_, hydrogen peroxide; GNT, gentamicin; VNC, vancomycin; AMX, amoxicillin; TCZ, tioconazole; nt, not tested; nd, not detected.

## Data Availability

Data are contained within the article.

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
