# Peer review of "Eco-Friendly Sanitization of Indoor Environments: Effectiveness of Thyme Essential Oil in Controlling Bioaerosol Levels and Disinfecting Surfaces"

_biotech, 2024, doi:10.3390/biotech13020012_

Round 1

Reviewer 1 Report

Comments and Suggestions for Authors

The article entitled "Eco-friendly Sanitization of Indoor Environments: Effectiveness of Thyme Essential Oil in Controlling Bioaerosol Levels and Disinfecting Surfaces" is a manuscript that clearly and comprehensively provides the data relating to a study on thyme essential oils.

Below I report a series of observations/modifications to make in order to improve the quality of your manuscript.

-Change the acronym SARS-CoV-19 with SARS-CoV-2.

-Please provide resistance/sensitivity cutoff values for the various in vitro study methods used.

-It would be appropriate to provide images of the results of the various in vitro study methods.

- Indicate the details and methods of the surface sanitization methods (paragraph 2.7 Indoor Environmental Surfaces Sanitization).

- Indicate why a swab with sterile sodium chloride solution was used for surface sampling (line 227). Does sodium chloride block the action of TEO?

- Indicate the limitations of the study in discussions.

-Provide more information on the studies already present in the literature on essential oils, in particular on TEO.

Reviewer 2 Report

Comments and Suggestions for Authors

The manuscript presents some interesting results on the use of thyme essential oil (TEO) for sanitization of surfaces and air, as well as antimicrobial activity tests.  However, there are a number of uncertainties and other issues that could be improved to make the work more impactful and insightful.  These are listed below in no particular order.

1. The use of "eco-friendly" in the title,  discussion about TEO being a natural substance, and suggestion that it is safe and preferable to chemical sanitizers is a bit problematic.  There are a number of papers in literature describing the potentially negative impacts on indoor air quality through the use of essential oils and other compounds, including some cited in this manuscript [6, 8].  The introduction of additional volatile organic compounds can raise the fine particulate concentrations in indoor air through the generation of secondary organic aerosols via complex reactions.  The manuscript would benefit from a bit more careful discussion of the potential benefits and limitations of using TEO, since the impact on air quality is still uncertain.

2. Line 95:  "inox" is more commonly referred to as "stainless steel" in some regions, so this might be clarified.

3. The specific choices of other sanitizers (hypochlorite and peroxide) is understandable but introduces some limitations in the agar well diffusion tests.  Both of these are reactive and somewhat unstable, and the use of something more similar to pure TEO might have been interesting.  Quaternary ammonium compounds. ethanol or a pure terpene could be other choices.  

4.  Discussion of the agar well diffusion results is somewhat complicated by the different chemical nature of the pure TEO and the aqueous solutions of the positive controls and other sanitizers, since the concentrations and diffusion rates can differ.  Some discussion of the limitations of this testing would be beneficial.  It would also be helpful to make more comparisons with other literature that has examined the antimicrobial activity of TEO, at least in a general way even if the methods are somewhat different.

5. In the disk volatilization testing, the hypochlorite and peroxide positive controls are not really that useful or relevant, since neither compounds have much volatility.  The use of something more volatile, for example ethanol, would have been more interesting.

6.  Methodology Sections 2.4 to 2.8 lack quite a few details that are necessary for replication or full comprehension of the situation.  

6.1:  some details of the test room are lacking.  For example, were the lights on during the testing time?  Was the HVAC system running, and if so what was the air exchange rate in the room?  This will have a significant impact on the concentrations of sanitizers in the air during nebulizer experiments.

6.2:  how long was the nebulizer operated to dispense the 100 mL of sanitizers?  The length of time these operate, together with the rate of air changes in the room (if any) will significantly affect the concentration of sanitizers in the air during the experiments.  Was sampling conducted immediately after the 100 mL was dispersed, or some time later?  Please clarify if the peroxide solution was also dispersed for 100 mL and similar times.

6.3:  for the settle plate bioaerosol sampling, how many sampling points were actually used?  Were they the same number and location for every test?  The settle plate technique can be inherently noisy, but it seems that good statistical analysis was used to assess the overall effects.  Were any location-dependent results obtained?  These might be of interest.

6.4:  for the surface sanitization, how long was the sanitizer allowed to contact the surface, and was this time always consistent?  Did the surface dry before sampling? Was the surface subsequently wiped with any cloth or towel, and if so was this also done to the control surface without sanitizer?  Did the sampling employ any neutralizer to counteract residual sanitizer that may be picked up during swabbing, or was this potential effect tested in preliminary work and found to be insignificant?  

6.5:  for surface swabbing, it is a bit unclear whether the same exact area was sampled before and after sanitization, or whether these were perhaps adjacent areas.  If they were the same area, the initial swabbing could remove microbes and affect the subsequent sampling performed after sanitization.

7. Around line 409:  the antimicrobial mechanisms of TEO are described, which largely depend on membrane effects.  This is why I raise the question of other controls such as ethanol, which has similar membrane disruption effects, versus peroxide and hypochlorite.  

8.  A comparison or discussion of TEO concentrations in the air versus those of peroxide would be more useful around line 455.  Why were the specific concentrations of 2.5% and 5% chosen for TEO?  Presumably, 5% TEO has a 10 x higher concentration in air versus 0.5% peroxide, assuming the nebulization and ventilation rates were consistent between the tests.  The actual TEO or thymol concentration in the air would be of significant interest, for comparison with other volatile compound (VOC) concentrations typically found in indoor air.  As noted above, adding more VOCs to the air has its own environmental concerns, depending on the complex atmospheric (indoor) reaction chemistry of each compound.

9.  A bit clearer description of what is new or novel in this work would be useful.  A lot of tests were performed, but there is also other work in literature on essential oils and antimicrobial activity.  Some clearer comparisons and contrasts with previous work would help.

Comments on the Quality of English Language

The English language quality is very good in general, with just a few minor errors.  For example a typo in line 326, and line 328 is awkward.

Round 2

Reviewer 2 Report

Comments and Suggestions for Authors

The authors have made some significant improvements to the description of methods, and these are much clearer now.  The additional discussion on the limitations of the use of these oils is a nice addition and helps to put the manuscript in more context and balance.  I thank the authors for their efforts.

One last comment:  On line 489, I think that "EO" should be "TEO"?